# Alteration of Autophagy and Glial Activity in Nilotinib-Treated Huntington’s Disease Patients

**DOI:** 10.3390/metabo12121225

**Published:** 2022-12-06

**Authors:** Karen E. Anderson, Max Stevenson, Rency Varghese, Michaeline L. Hebron, Erin Koppel, Mara McCartin, Robin Kuprewicz, Sara Matar, Dalila Ferrante, Charbel Moussa

**Affiliations:** 1MedStar Georgetown University Hospital Huntington’s Disease Research, Education and Care Center, Department of Psychiatry and Department of Neurology, Georgetown University Medical Center, 4000 Reservoir Rd, NW, Washington, DC 20057, USA; 2Translational Neurotherapeutics Program, Laboratory for Dementia and Parkinsonism, Department of Neurology, Georgetown University Medical Center, Building D, Room 265, 4000 Reservoir Rd, NW, Washington, DC 20057, USA; 3Genomics and Epigenomics Shared Resource, Department of Oncology, Georgetown University Medical Center, Building D, 4000 Reservoir Rd, NW, Washington, DC 20057, USA

**Keywords:** nilotinib, Huntington’s, DDR1, dopamine, miRNA, autophagy, inflammation, basal ganglia

## Abstract

Nilotinib is a tyrosine kinase inhibitor that is safe and tolerated in neurodegeneration, it achieves CSF concentration that is adequate to inhibit discoidin domain receptor (DDR)-1. Nilotinib significantly affects dopamine metabolites, including Homovanillic acid (HVA), resulting in an increase in brain dopamine. HD is a hereditary disease caused by mutations in the *Huntingtin’s* (HTT) gene and characterized by neurodegeneration and motor and behavioral symptoms that are associated with activation of dopamine receptors. We explored the effects of a low dose of nilotinib (150 mg) on behavioral changes and motor symptoms in manifest HD patients and examined the effects of nilotinib on several brain mechanisms, including dopamine transmission and gene expression via cerebrospinal fluid (CSF) miRNA sequencing. Nilotinib, 150 mg, did not result in any behavioral changes, although it significantly attenuated HVA levels, suggesting reduction of dopamine catabolism. There was no significant change in HTT, phosphorylated neuro-filament and inflammatory markers in the CSF and plasma via immunoassays. Whole miRNA genome sequencing of the CSF revealed significant longitudinal changes in miRNAs that control specific genes associated with autophagy, inflammation, microglial activity and basal ganglia neurotransmitters, including dopamine and serotonin.

## 1. Introduction

Huntington’s disease (HD) is a hereditary autosomal dominant disease caused by expansion of a cytosine-adenine-guanine (CAG) triplet-repeat within the first exon of the *Huntingtin* (*HTT*) gene [1]. The primary neuropathology in HD is loss of striatal and cortical neurons [2]. Post-mortem studies indicate that HD is characterized by degenerative changes in striatal medium spiny neurons (MSNs) [3,4,5]. HD causes uncontrolled movements, including involuntary jerking or twitching known as chorea [6], psychiatric symptoms and cognitive decline [7,8] and changes in personality, i.e irritability [9]. The role of dopamine (DA) is critical in HD, and DA antagonists are effective treatment for motor and behavioral symptoms [10]. Abnormal DA transmission or activation of DA D1 and D2 receptors may contribute to motor and behavioral symptoms in human [11] and animal models of HD [12,13,14]. Mutant *HTT* is also associated with decreased DA levels [15].

Nilotinib is a tyrosine kinase (TK) inhibitor that potently inhibits Discoidin Domain Receptors (DDRs), including DDR1 with half-maximal inhibitory concentration (IC_50_) of 1 nM [16,17,18,19,20]. Treatment of Parkinson’s patients (PD) with nilotinib, 150 mg or 300 mg, results in maximum cerebrospinal fluid (CSF) concentration (C_max_) of 1.9 nM and 4.12 nM, respectively, and a dose-dependent increase in DA levels [21,22,23]; achieving a pharmacologically adequate concentration that would inhibit DDR1. Nilotinib is FDA-approved for the treatment of Chronic Myelogenous Leukemia (CML) because it also inhibits Abelson TK (IC_50_ > 20 nM) at higher (20×) concentration. Importantly, collagen activation of DDR1 [24] changes the activity of microglia and alters matrix metalloproteases (MMPs), resulting in disintegration of blood-brain-barrier (BBB) and inflammation [25]. Deletion or inhibition of DDR1 attenuates neuro-inflammation and improves CNS autophagy, bioenergetics and vesicular transport [17,26,27,28,29]. Significant alterations in miRNAs that control autophagy and inflammation genes are observed in PD progression, while nilotinib, targets DDR1 and reverses these miRNA levels, in agreement with the preclinical effects of nilotinib on the molecular pathways of autophagy and inflammation [21,22,23,29,30,31,32,33]. Previous studies in PD and Alzheimer’s disease (AD) demonstrated that nilotinib reduces brain DA breakdown as measured by DA metabolites, including homovanillic acid (HVA) [22,23,34], and it inhibits mono-amine oxidase (MAO)-B activity in a dose dependent manner [34]. The current study examined the effects of the lowest available dose of nilotinib, 150 mg, and determined whether it is safe and tolerated in individuals with manifest HD. The primary objective of this study was to understand if nilotinib effects on DA can lead to exacerbation of chorea and psychiatric symptoms (e.g., irritability) and measure HVA, phosphorylated neurofilaments, unbiased whole genome CSF miRNA sequencing and HTT levels as biomarker objectives at baseline and 3 months. Exploratory clinical outcomes included cognitive, motor and psychiatric symptoms at baseline, 3 months and 4 months wash-out period.

## 2. Results

### 2.1. Nilotinib Is Safe and Tolerated in HD Patients

Approximately 15 subjects were approached, 10 were screened, 4 did not meet inclusion criteria and 6 were enrolled and included male and female (2:1) with average age 55.33 ± 6.18 (year ± SD). There were no serious adverse events (SAEs) and no dropouts. Nilotinib, 150 mg, did not worsen or induce chorea and did not result in behavioral changes. There were 19 AEs (Appendix A) in this study and the most common AEs were skin disorders (36.84%) followed by less common neurological disorders, including headache (15.8%), but none of these AEs were reported as post-LP headaches. Two patients reported psychiatric symptoms (10.5%), including one patient who had increased irritability (5.2%), which seemed to increase after baseline and decrease after 3 months during wash out period. Another patient had obsessive compulsive disorders (OCD) only during the wash out period (5.2%). All other AEs were rare (<10%). There was no QTc prolongation (Appendix A) in any patient at throughout any visits.

### 2.2. Measurement of DA Metabolites and HTT

Measurement of DA metabolite HVA showed a non-significant decrease in CSF (Figure 1A, 46%) and a non-significant increase in plasma (Figure 1B, 131%), but the ratio of CSF: plasma HVA was significantly reduced (Figure 1C, 58%, *p* < 0.05, *n* = 5), suggesting reduction of DA catabolism. Measurement of total HTT levels did not change in CSF (Figure 1D, 104%) and plasma (Figure 1E, 95%), and the ratio of CSF:plasma HTT did not change (Figure 1F, 109%) The levels of phosphorylated neurofilaments did not change in the CSF (Figure 1G) and plasma (Figure 1H).

### 2.3. CSF miRNAs Expression Are Differentially Altered at Baseline and 3 Months in HD

To assess whether miRNAs are altered after 3 months nilotinib treatment, we performed whole genome miRNA sequencing in the CSF (*n* = 5) collected at baseline and 3 months from each individual patient (Figure 2A), to allow us to measure longitudinal miRNA changes. miRNAs inversely regulate mRNA, and they do not necessarily require large changes to affect their targets, therefore, we assessed all miRNA that met *p* < 0.05 between 3 months and baseline. Unique Molecular Identifier (*UMI*) counts for 2681 miRNA were mapped (Figure 2A,B) to the human genome in all samples, suggesting detection and sequencing of the entire CSF miRNA genome. The difference in absolute *UMI* counts between 3 months and baseline were plotted and hierarchical clustering was performed using Morpheus (Broad Institute, Boston MA) online tools (Figure 2B). Longitudinal expression analysis to determine differentially expressed miRNA (DEM) were performed in parallel using DESeq2 (Bioconductor) for ratio of means normalization followed by Wald testing and NOIseq (Bioconductor) for upper quantile normalization. 102 DEMs (*p* < 0.05) were found to be altered between 3 months and baseline in HD patients.

Gene Ontology (GO) analyses (Figure 2B,C) demonstrated that 69 miRNAs were found to be commonly altered among genes that regulate three major pathways, including autophagy (80 miRNAs), immunity (93 miRNAs) and neurodegenerative diseases (NDDs, 87 miRNAs). Of these commonly shared 69 miRNAs, miRNAs that control AKT2 (hsa-miR-6794-5p), BCL2 (hsa-miR-1-3p, hsa-miR-4262), FOS (hsa-miR-4267), mTOR (hsa-miR-496) MAPK3 (hsa-miR-4267), CREB1 (hsa-miR-203a-3p), MYD88 (hsa-miR-1909-3p), and TNF (hsa-miR-1343-5p) are all upregulated, suggesting suppression of target genes. On the other hand, more miRNAs targeting genes including CHP1 (hsa-miR-3134, hsa-miR-644a), HSPA9 (hsa-miR-3934-3p), MAPK1 (hsa-miR-454-5p), PIK3CD/G (hsa-miR-6773-5p, hsa-miR-378b, hsa-miR-6829-3p), PPP3R1/2 (hsa-miR-5187-3p, hsa-miR-3925-3p), PRKCQ (hsa-miR-6840-3p), and VEGFA (hsa-miR-2115-3p, hsa-miR-302c-3p) are all downregulated suggesting increased expression of target genes (Figure 2C).

### 2.4. miRNAs That Control Neurotransmitters of Basal Ganglia Pathways Are Significantly Altered

microRNA Target were matched with associated mRNAs or targets that are reported or already verified experimentally using Ingenuity Pathway Analysis (IPA, QIAGEN Inc., Rockville, MD, USA). Experimentally validated interactions from TarBase, miRecords, as well as highly predicted miRNA-mRNA interactions from TargetScan are identified. Specific miRNAs that control DA, serotonin and other neurotransmitters of basal ganglia pathways are significantly differentially expressed at 3 months-baseline (Figure 3). Upregulated were miRNAs that target CALM1 (hsa-miR-3616-3p) CAMK4 (hsa-miR-4531, hsa-miR-4646-5p), DDC (hsa-miR-4531), DRD2 (hsa-miR-939-5p), GCH1 (hsa-miR-1-3p, hsa-miR-133a-3p), KCNJ10 (hsa-miR-1233-5p, hsa-miR-1275) NOS1 (hsa-miR-6794-5p) PLCD3 (hsa-miR-6813), and TH (hsa-miR-1909-3p) among others, while downregulated were miRNAs that target ALDH1B1 (hsa-miR-6796-5p), CSNK1E (hsa-miR-219b-5p), GNAS (hsa-miR-656-5p), GRIN2B (hsa-miR-153-5p, hsa-miR-181b-3p), MAOB (hsa-miR-4322), NCS1 (hsa-miR-6794-5p), PTS (hsa-miR-4432), SPR (hsa-miR-920, hsa-miR-939-5p), and SULT1B1 (hsa-miR-95-5p), among others, genes predicted by IPA to regulate DA receptor signaling and degradation. Additionally, upregulated were miRNAs targeting various HTR subunits (hsa-miR-1275, hsa-miR-153-5p, hsa-miR-4646-5p), ADCY4 (hsa-miR-4674), LARGE1 (hsa-miR-656-5p), and SLC18A2 (hsa-miR-125b-2-3p), while downregulated were miRNAs targeting ADH4 (hsa-miR-7153-3p), GUCY1A1/B1 (hsa-miR-302c-3p, hsa-miR-219b-5p), PCBD1 (hsa-miR-4693-3p), and UTG1A3 (hsa-miR-6839-3p), genes predicted by IPA to regulate serotonin receptor signaling and degradation.

### 2.5. Longitudinal Alterations of miRNAs That Regulate Autophagy in HD

Key autophagy genes *ATG9B* (hsa-miR-920), *ATG12* (hsa-miR-3064-3p), *BECN1* (hsa-miR-3675-3p, hsa-miR-4797-3p), and *SESN1* (hsa-miR-6715b-3p) are gene targets associated with autophagy as predicted using Ingenuity Pathway Analysis (IPA) software (Figure 4). The miRNAs targeting these genes are significantly decreased over 3 months following nilotinib treatment, suggesting an increase in the expression of these genes and amelioration of autophagy. These genes are well characterized for their roles in the formation and initial construction of the autophagosome for macro-autophagy and chaperone-mediated autophagy [35]. We observed a significant increase of miRNAs associated with regulation of *BCL2* (hsa-miR-1-3p, hsa-miR-4262) and *ATG5* (hsa-miR-879-5p, hsa-miR-766-5p), suggesting autophagosome maturation (Figure 4). Together, an increase in *ATG12* expression and decrease in *MAP1LC3A/B* (hsa-miR-1307-5p, hsa-miR-1233-5p) expression indicate autophagosome clearance [28,29,33,35]. Increased expression of *MAPLC3C* (hsa-miR-4252) indicates accumulation of autophagosomes [28,29,33,35], but the absence in the differential expression of this gene at baseline-3 months may be related to autophagosome clearance. On the other hand, expression of hsa-miR-5195-3p, which targets *LAMP2* expression, was reduced, indicating that LAMP2 expression is increased to mediate maturation of the autophagosome [36,37]. Loss of LAMP2 leads to phagophore accumulation and blockade of autophagy flux [36,37]. Taken together these data indicate that nilotinib, 150 mg, may influence autophagy in HD patients. Other miRNAs that simultaneously control genes involved in autophagy and other molecular mechanisms are discussed with their respective pathways. Overall, this pattern of disinhibited and suppressed miRNAs and the effects on their respective genes suggest neuronal activity to complete autophagy flux [35], consistent with animal data [28,29] and post-mortem human brains [38,39] showing that nilotinib treatment facilitates clearance of neurotoxic proteins via autophagy. We observed disinhibition of *SQSTM1* (hsa-miR-3137), which mediate substrate recognition and recruitment for lysosomal degradation. Alterations of autophagy-ubiquitination genes are consistent with significant changes of vesicular transport (VPSs) genes (hsa-miR-1243) indicating facilitation of cellular transport [26] and protein clearance.

### 2.6. Alterations of miRNAs Targeting Genes That Regulate Sirtuin Pathways and Microglial Activity

Nilotinib, 150 mg significantly altered energy metabolism, including the Sirtuin pathways (Figure 5), as evidenced by changes in expression of miRNAs that target SIRT1 (hsa-miR-6715b-3p), SIRT2 (hsa-miR-1275), SIRT5 (hsa-miR-5089-3p), SIRT6 (hsa-miR-1909-3p), and MYC (hsa-miR-3934-3p) among others, suggesting involvement of the sirtuin pathways. Additionally, upregulated were miRNAs targeting such genes as SOD3 (hsa-miR-6794-5p), TNF (hsa-miR-125b-2-3p, hsa-miR-656-5p, hsa-miR-939-5p), BAX (hsa-miR-766-5p), and FOXO3 (hsa-miR-1909-3p), genes associated with cellular senescence and apoptosis, while downregulated were miRNAs targeting genes such as ATG12 (hsa-miR-4295, hsa-miR-155-5p), BECN1 (hsa-miR-30a-5p), RELA (hsa-miR-302c-3p) and MAPK1 (hsa-miR-454-5p), genes associated with anti-oxidation and clearance of intracellular waste. Additionally, we found enrichment of gliogenesis (hsa-miR-454-5p, hsa-miR-4267, hsa-miR-3934-3p, hsa-miR-1275, hsa-miR-302c-3p) (Figure 5). There were also changes in several neuro-inflammatory markers, including Interleukins (hsa-miR-4638-5p, hsa-miR-3973, hsa-miR-4638-5p, hsa-miR-4797-3p, hsa-miR-920) TNF (hsa-miR-125b-2-3p, hsa-miR-656-5p, hsa-miR-939-5p), IFN-gamma (hsa-miR-1243) NFkB (hsa-miR-892b) and markers for Th1 and Th2 lymphocyte proliferation such as NOTCH (hsa-miR-1-3p, hsa-miR-4262, hsa-miR-4322), NFAT (hsa-miR-133a-3p), TGFB3 (hsa-miR-6886-33p), TLRs (hsa-miR-4655-5p, hsa-miR-4634, hsa-miR-1909-3p) and CD4 (hsa-miR-939-5p), and CD40 L (hsa-miR-181b-3p, 6794-5p) as shown in Appendix A, suggestive of molecular regulation of neuro-inflammatory pathways and microglial activity. Furthermore, ELISA measurement of several inflammatory molecules that regulate microglial activity (Appendix A), including reduction of inflammatory IL-5 an increase of anti-inflammatory IL-6 as well as MCP3 that control microglial activity and release of inflammatory factors [40] showed a trend towards microglial modulation.

### 2.7. Alterations of miRNAs Targeting Genes Associated with Neurodegenerative Disease Processes

Concurrent with the changes in expression of miRNAs targeting genes associated with autophagy, sirtuin signaling, and inflammation, we also detected alterations in miRNAs that interact with genes reported to be associated with various neurodegenerative diseases that feature aggregation of neurotoxic proteins. Following treatment with nilotinib, we identified significant changes in expression of miRNAs that interact with SOD3 (hsa-miR-6794-5p), APP (hsa-miR-302c-3p), presenilin (hsa-miR-1233-5p, hsa-miR-6829-3p), BACE1 (hsa-miR-4531), BDNF (hsa-miR-1-3p), CYCS (hsa-miR-3925-3p, hsa-miR-454-5p, hsa-miR-6797-3p), TCERG1 (hsa-miR-4262), and HDAC (hsa-miR-4634, hsa-miR-6794-5p, hsa-miR-3189-3p, hsa-miR-326), genes associated with pathogenesis in such diseases as AD, PD, and Amyotrophic Lateral Sclerosis (ALS), in addition to HD, as shown in Appendix A. These miRNAs also target pathways associated with processes such as mitochondrial dysfunction (hsa-miR-1233-5p, hsa-miR-1275, hsa-miR-1301-3p, hsa-miR-181b-3p, hsa-miR-186-3p) oxidative stress (hsa-miR-1909-3p, hsa-miR-200b-5p, hsa-miR-203b-5p, hsa-miR-2115-3p), and superoxide radical degradation (hsa-miR-3064-3p, hsa-miR-656-5p, hsa-miR-6829-3p) features described in these diseases as promoting neurodegeneration. This suggests that treatment with nilotinib may be altering the activity of these pathways as a means of mitigating their neurodegenerative properties.

### 2.8. Exploratory Clinical Effects

We compared baseline with the effects of 3-month nilotinib treatment. No changes in exploratory cognitive outcomes including MoCA and TMT-B, motor endpoints UHDRS and TUG and psychiatric assessments, including PBA-s and IAS were observed in this study (Appendix A).

## 3. Discussion

This study primarily evaluated the effects of nilotinib, 150 mg, on safety and tolerability in patients with HD. All patients tolerated the treatment with no AEs, including QTc prolongation, myelosuppression, hepatotoxicity, and pancreatitis. The safety and tolerability of nilotinib were also observed in several clinical studies in patients with neurodegeneration, including PD [21,22,23,34] and AD [41]. Prior studies showed that nilotinib treatment consistently alters the levels of DA metabolites, including HVA, in the CSF and plasma reflecting a reduction of DA catabolism and elevation of DA levels [21,22,23,34,41]. Nilotinib treatment increased DA levels and led to stabilization of CSF alpha-synuclein levels and long-term motor symptoms in PD patients [21,22,23,34]. This study evaluated the effects of potential changes of DA levels on chorea and behavioral symptoms, including irritability, and demonstrated no worsening in motor and behavioral symptoms in manifest HD. Two patients reported psychiatric symptoms, including one patient who had been irritable prior to enrollment but continued to display slight worsening after baseline, suggesting that this behavioral worsening may be due to nilotinib effect and/or DA increase or disease progression. Another patient displayed OCD only during the wash out period. Taken together, these data indicate that a low dose of 150 mg nilotinib is safe and may not induce behavioral and motor symptoms in manifest HD patients. However, this study is very small and explorative (phase 1b) and lacks a placebo group, suggesting that increased DA levels may lead to activation of DA D1 and D2 receptors and contribute to motor and behavioral symptoms as is observed in human [11] and animal models of HD [12,13,14].

Dopaminergic transmission may exert some long-term neuroprotective effects different than their role in managing some symptoms of HD. Activation of DA receptors modulate adenylyl cyclase (AC) and trigger signal pathways of neurodegeneration in HD [42]. For example, D1-like DA receptors family (D1Rs) couple positively to AC to trigger accumulation of intracellular cyclic 3,5 adenine-monophosphate (cAMP) and subsequent activation of the protein kinase dependent of cAMP (PKA). In contrast, the D2-like dopamine receptors family (D2Rs) negatively couple to AC, which results in decreases of cAMP accumulation [43], and regulation of the activity of PKA and DARPP-32 (dopamine- and cAMP-regulated neuronal phosphoprotein)/protein phosphatase-1 cascade [44]. Pridopidine, a DA stabilizer, could improve motor performance and result in neuroprotection in R6/2 mouse model of HD [45], upregulate the BDNF pathway and sigma 1 receptor (S1R) in Q175 knock-in (Q175 KI) vs. Q25 WT mouse models of HD [46]. DA stabilization via pridopidine also upregulates expression of AKT/PI3K pathways, which promote neuronal plasticity and survival [46]. Recent evidence suggest that Pridopiline is a sigma-1 and DA D2/D3 receptor agonist in healthy volunteers and HD patients [47]. Furthermore, altered expression of neurotransmitter receptors, including DA, serotonin, and glutamate, may precede clinical symptoms in R6/2 mice and may contribute to subsequent pathology [48,49]. We observed significant changes in basal ganglia neurotransmission at 3 months compared to baseline, suggesting that the change in DA, serotonin and other neurotransmitter metabolism may influence basal ganglia, leading to motor control over a longer period of nilotinib treatment. Interestingly, HTT is a substrate of the Akt pathway that is activated by insulin growth factor (IGF)1 [50], which was significantly observed in our study. Reduced activation of the Akt pathway decreases phosphorylation of mutated HTT, leading to neuronal toxicity [51]. Low levels of IGF1 might also alter Akt activity that is observed in HD animal models and HD patients [52]. Mutant *HTT* is also associated with decreased DA levels [15], whereas definitive results in HTT levels could be shown due to small sample size and/or short duration of treatment. Taken together, a larger placebo-controlled study will enable us to characterize the effects if these important genes and pathways in HD patients treated with nilotinib.

Nilotinib is a potent inhibitor of DDR1 (IC_50_ 1 nM) [16,17,18,19,20,53] and it adequately enters the brain in PD (up to 4.12 nM) [21,22,23,34] and AD (up to 4.7 nM) [41] patients to inhibit DDR1. Importantly, DDR1 activation alters microglial activity [25], whereas deletion or inhibition of DDR1 attenuates neuro-inflammation and improves autophagy, bioenergetics and vesicular transport in the CNS [17,26,27,28,29]. We previously observed significant alterations in miRNAs that control autophagy and inflammation genes in PD patients treated with nilotinib, which reverses these miRNA levels [53] in agreement with the preclinical effects of nilotinib on the molecular pathways of autophagy and inflammation [21,22,23,29,30,31,32,33]. Nilotinib, 150 mg and 300 mg, led a dose dependent reduction of neurotoxic proteins, including tau in PD [22] and amyloid in AD [41] patients. These effects of nilotinib suggest a disease modifying effect in neurodegeneration, thus we examined the effect of a low dose of nilotinib on gene expression in HD via whole genome sequencing of CSF miRNA. miRNAs post-transcriptionally regulate gene expression, primarily via silencing of their target mRNA. A number of miRNAs that control genes associated with ubiquitination and autophagy, inflammation, basal ganglia neurotransmitters, Sirtuin and bioenergetics were identified in the CSF of HD patients [26], and these miRNA were, consistent with the impairment of autophagy in post-mortem human brain [39,54]. miRNAs are dynamic regulators of gene expression and maybe important indicators of gene expression in this small cohort and short-term treatment in this study. We observed that nilotinib treated HD patients display significant directional changes of miRNAs that control genes of neuro-inflammation and microglial activity, improvement of CNS autophagy, bioenergetics and vesicular transport [17,26,27,28,29]. Several pathways by which mHTT may cause cell death have been identified [55], including effects on chaperone proteins (SQSM1), autophagy, impairment of energy metabolism; and neuroinflammation [55,56]. These pathways were shown to be affected in manifest HD patients but it is unclear whether the changes observed in miRNAs are due to nilotinib treatment or disease progression. Nonetheless, a longer duration of treatment in a placebo-controlled larger phase 2 study may reveal the effects of nilotinib on the neuropathology of HD.

There is compelling evidence to support regulation of SIRT1 and SIRT2 that are shuttled between the nucleus and cytoplasm [57] and their inhibition might be beneficial in neurodegenerative diseases, including HD [57,58]. SIRT miRNAs appeared to be a major pathway that is regulated in this study, however, it remains unclear whether changes in SIRT are beneficial or detrimental. Sirt1 inhibitor selisistat can suppress HD pathology caused by mHTT, but genetic depletion of Sirt2 reverses the effect of this inhibitor [59]. The sirtuin pathway is also linked to longevity, energy metabolism and mitochondrial integrity. Our miRNA analysis supports prior findings that show deficits in energy metabolism, fatty acids metabolism, mitochondrial TCA and oxidative phosphorylation in HD patients, including pre-manifest and early HD, and animal models (review) [60]. Striatal hypometabolism, as measured by Fluorodeoxyglucose [^18^F] (^18^F-FDG) positron emission tomography (PET) is associated with clinical disease severity in HD [48] and systemic metabolic and energy defects were demonstrated in HD [61].

We previously demonstrated that changes in miRNAs that are associated with growth factors and angiogenesis in nilotinib [53] are also associated with a significant increase in DA levels [22,23,34,41]. DDR1 attenuates inflammation in models of atherosclerosis [62] we previously showed that vascularization could facilitate transport across the BBB and result in a beneficial long-term immune response and potential clinical stabilization [34,53]. Partial or complete knockdown, or pharmacological inhibition of DDR1, attenuates neuro-inflammation and improves CNS autophagy and vesicular transport [17,26,27,28,29], as suggested by miRNA analysis in HD patients. Microglia activity and inflammatory markers such as IL-6 are elevated in the CSF of early HD patients [63,64], consistent with our data. Overall, changes in miRNAs that control autophagy genes reduces expression of proteins that mediate ubiquitination and the endosomal and autophagy-lysosomal pathways, [26] may be concurrent with changes in miRNAs that control inflammation and energy metabolism, consistent with previous data that show nilotinib attenuates or reverses the level of miRNA expression that control these pathways in placebo-controlled PD trials [26,53] and animal data [29,30,31,32,33].

In conclusion, nilotinib, 150 mg, is safe and tolerated in HD and does not exacerbate motor and behavioral symptoms. The miRNA studies led to detection of the complete miRNA genome in the CSF and may provide a useful tool to study disease-modifying effects such as autophagy, inflammation, and energy metabolism in manifest HD patients. Taken together, this study provides proof-of-concept to conduct a larger phase 2 study with longer treatment period (6–12 months) and perhaps a dose range of nilotinib to determine its effects on HD.

## 4. Materials and Methods

### 4.1. Study Design

This an open label, Phase Ib, proof of concept study to evaluate the impact of a low dose (150 mg) of nilotinib treatment on safety, tolerability, and potential biomarkers in participants with manifest HD who were not receiving any MAO inhibitors or any other DA related treatments. A total of 6 participants were enrolled and received an oral dose of 150 mg nilotinib once daily for 3 months, followed by 1-month wash-out. Our primary objectives were to investigate whether HD patients tolerate nilotinib, with no exacerbation of chorea and behavioral symptoms and no other Adverse Events (AEs) such as myelosuppression, QTc prolongation, liver/pancreatic toxicity, etc. This study was conducted in subjects in HD patients with Total Functional Capacity (TFC) of 7–12, indicating early to moderate disease, and Montreal Cognitive Assessment (MoCA) score ≥16 to identify mild cognitive impairment (MCI) at screening. Screening and enrollment of all participants were from MedStar Georgetown Hospital (GUH) Huntington’s Disease Research, Education and Care Center. We compared baseline with the effects of 3-month nilotinib treatment. All participants who met all inclusion/exclusion criteria had 2 mandatory lumbar punctures (LPs) at baseline and 3 months end of treatment. Biomarker objectives included measurement of HVA levels, HTT, neurofilaments and markers of neuro-inflammation. CSF was also collected to perform next generation whole genome sequencing at baseline and 3 months. Exploratory outcomes included clinical cognitive assessments via MoCA and Trail Making Test B (TMT-B). Motor assessments were performed via the Unified Huntington’s Disease Rating Scale-motor (UHDRS) and Timed-Up and Go (TUG) and psychiatric assessments via Problem Behaviors Assessments Short Form (PBA-s) and the Irritability and Apathy Scale (IAS). A follow up 4 months visit included all study procedures, except LP. No formal power analysis was performed as this is an exploratory proof-of-concept study with a primary objective that the known effects of nilotinib on dopamine metabolism does not exacerbate behavioral symptoms (e.g., irritability) and chorea in a small population of HD.

### 4.2. Study Approval

The protocol was approved 12 July 2018 by the Institutional Review Board (IRB #2017-0440) at Georgetown University Medical Center as well as by the Georgetown-Howard Universities Center for Clinical and Translational Science (GHUCCTS) scientific review board. The study was listed on clinicaltrials.gov (NCT03764215) and Investigational New Drug (IND#139031). All participants were consented at the GHUCCTS Clinical Research Unit (CRU). Patients were de-identified, and a Pad-ID was included (e.g., NIL) in the patient’s clinical chart. An independent data safety and monitoring board (DSMB) that included a cardiologist, hematologist, clinical pharmacologist, neurologist, and biostatistician monitored safety signals in patients in the study.

### 4.3. Cerebrospinal Fluid (CSF) Collection and HVA Measurement

In total up to 20 mL of CSF was taken from 5 of the 6 subjects during this study using procedures we previously reported [22,23,34]. We were not able to obtain CSF on one subject using manual or fluoroscopy guidance. HVA measurements were performed via mass spectrometry as we previously reported [22,23,34].

### 4.4. Neuro-Inflammatory Markers Panel

We used multiplex Xmap technology that uses magnetic microspheres internally coded with two fluorescent dyes to measure markers of neurodegeneration. All samples were incubated overnight at 4 °C with 25 µL of a mixed bead solution, containing human neuro-inflammatory markers (Millipore, Cat#: HCYTA-60K-PX38) or HTT (MyBioSource, CAT#: 50-1560-2756). Samples were incubated with 25 µL detection antibody solution for 1.5 h at room temperature (Millipore, Cat#: HCYTA-60K-PX38), and 25 µL of Streptavidin-Phycoerythrin were used for detection. The Median Fluorescent Intensity (MFI) data was analyzed using a 5-parameter logistic or spline curve-fitting method for calculating analyte concentrations in samples.

### 4.5. microRNA Sequencing

CSF (200 µL) was used to isolate cell-free total RNA using the Qiagen miRNAeasy serum/plasma extraction kit (Qiagen, 217184) and CSF microRNA sequencing was performed as we previously explained [53]. Qiagen QiaSeq miRNA-seq library preparation kit (Qiagen, 331502 was used to normalize samples to an input volume of 5 µL RNA eluate and obtain miRNAseq libraries.) NextSeq 550 Sequencing System (Illumina) using single-end (SE) 1 × 75 base pairs (bp) sequencing to a depth of 25 million raw reads per sample was used to perform unbiased Next-generation sequencing (NGS). miRNA quantification was performed via uploading of FASTQ files to the online Qiagen Data Analysis Center. The unique molecular index (UMI) counts were calculated, and primary miRNA mapping was performed using a human-specific miRBase mature database. In the primary QIAseq quantification step, adaptor sequences from the library preparation process and any low-quality bases were removed. UMI counts for each miRNA were used for differential expression analysis.

### 4.6. microRNA Expression Analysis and Gene Ontology

A total of 2681 miRNA expression profiles at the baseline and 3 months were quantified in UMI counts., Longitudinal differentially expressed miRNA (DEMs) were determined via DESeq2 with ratio of means normalization and followed by Wald test [65] to compute the longitudinal measurement at baseline and 3 months. The DEMs were further evaluated using the Ingenuity Pathway Analysis (IPA, QIAGEN Inc., Rockville, MD, USA) microRNA Target Filter to match associated gene targets that are reported or already verified based on experimentally validated interactions from TarBase and miRecords, as well as predicted interactions from TargetScan. Gene targets were then filtered to include only highly predicted and experimentally supported mRNA target predictions. Gene targets were filtered to include only highly predicted and experimentally supported mRNA target predictions. Gene ontology (GO) pathway and functional enrichment analyses were performed on the gene targets of the significant DEMs using PANTHER Classification System [66]. Only enriched pathways and GO terms that met Benjamini-Hochberg *p* < 0.05 were reported.

### 4.7. Posthoc Power Analysis

When 5% of miRNAs (123 among approximately 2500 miRNA) are informative with log-2 or greater fold changes, a mean count of 500, and a common dispersion of 0.1, 5 samples within treatment arms would provide approximately 95% power at the false discovery rate (FDR) of 0.05 [67].

## 5. Patents

Charbel Moussa is an inventor on a Georgetown University (GU) US and International Patent to use Nilotinib in neurodegenerative diseases. GU exclusively licensed Nilotinib use patent to KeiffeRx. Charbel Moussa is co-founder and shareholder of KeiferX, and Charbel Moussa to KeifeRx.

## Figures and Tables

**Figure 1 metabolites-12-01225-f001:**
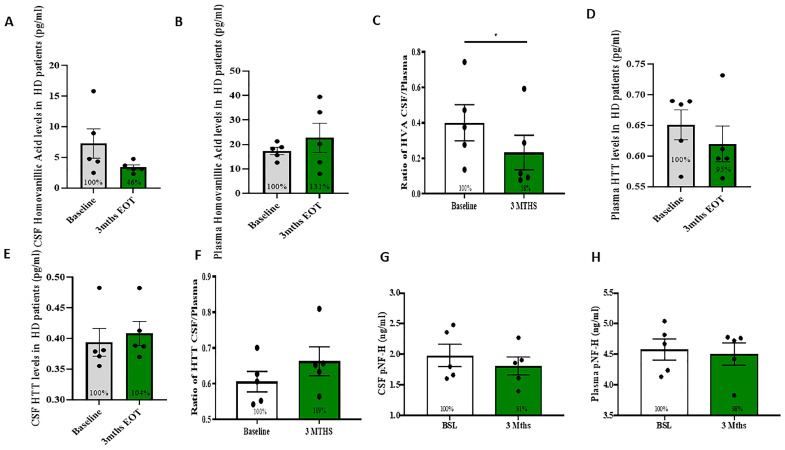
Measurement of potential biomarkers at baseline and 3 months following nilotinib treatment. ELISA levels of HVA in the (**A**) CSF and (**B**) plasma and (**C**) ratio of CSF/plasma HVA in HD patients. ELISA levels of HTT in the (**D**) plasma and (**E**) CSF and (**F**) ratio of CSF/plasma HTT in HD patients. ELISA levels of phosphorylated neurofilaments in the (**G**) CSF and (**H**) plasma in HD patients. *n* = 5, * designates *p* < 0.05.

**Figure 2 metabolites-12-01225-f002:**
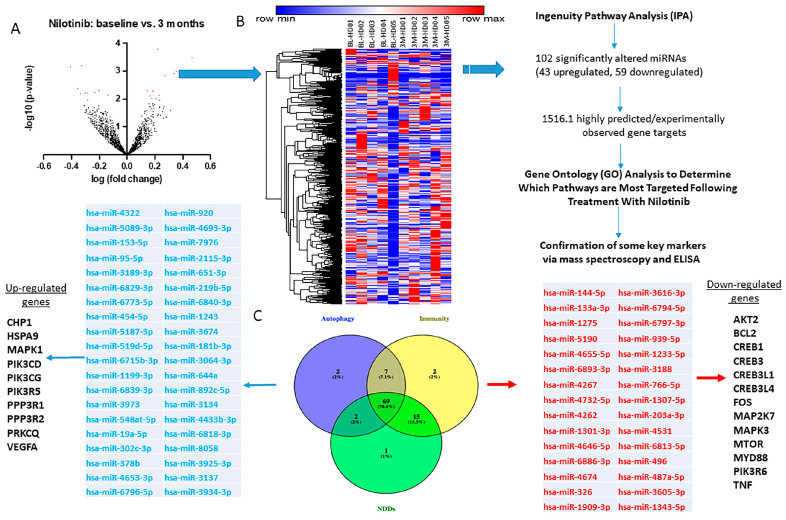
Treatment with nilotinib results in differential miRNA expression. (**A**) Volcano plot showing differentially regulated miRNAs following 3 months of treatment with nilotinib; miRNAs with a −log_10_
*p*-value > 2 are displayed in red (significance: *p* < 0,05). (**B**) Processing of sequenced miRNAs; miRNAs were analyzed to look for significant alterations in expression using Ingenuity Pathway Analysis (IPA), which revealed 102 distinct miRNAs that targeted 15,161 genes. (**C**) Gene ontology analysis revealed that autophagy, immunity, and neurodegenerative disease pathogenesis were the pathways most targeted by both up and downregulated miRNAs following nilotinib treatment, identifying a series of genes associated with these pathways that are targeted by selected miRNAs.

**Figure 3 metabolites-12-01225-f003:**
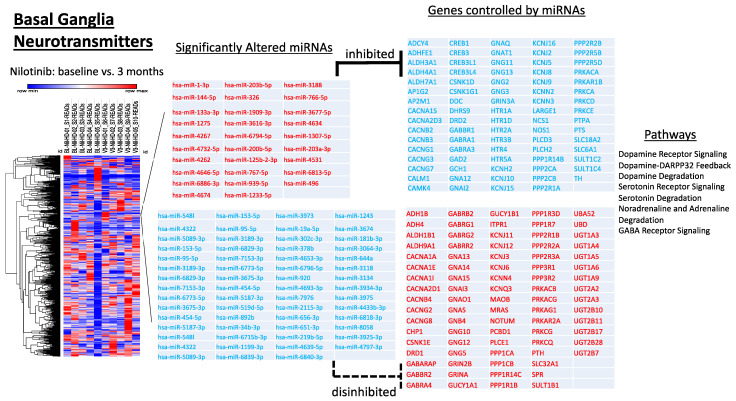
miRNAs target genes associated basal ganglia neurotransmitters. Gene ontology analysis revealed 29 upregulated and 59 downregulated miRNAs that target 162 genes experimentally observed to be associated with dopamine, serotonin, noradrenaline, adrenaline, and GABA signaling.

**Figure 4 metabolites-12-01225-f004:**
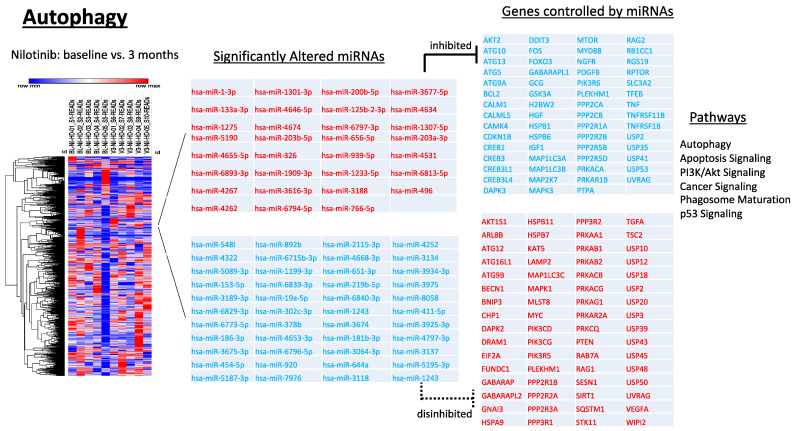
miRNAs target genes associated with autophagy. Gene ontology analysis revealed 31 upregulated and 44 downregulated miRNAs that target 122 genes experimentally observed to be associated with autophagic processing.

**Figure 5 metabolites-12-01225-f005:**
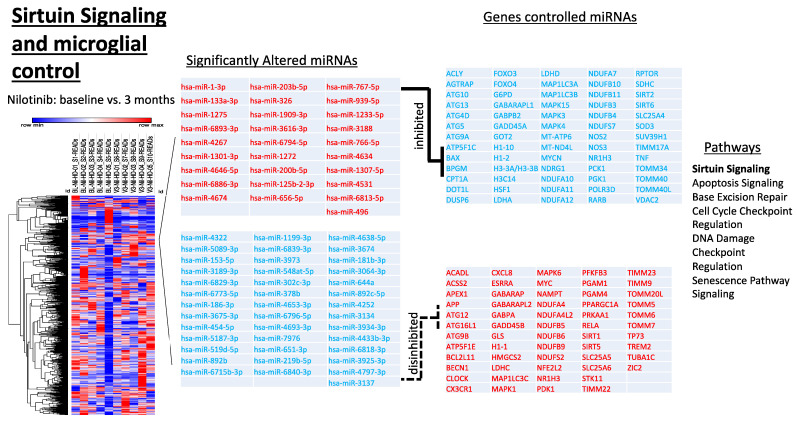
miRNAs targeting genes associated with sirtuin signaling and microglial control. Gene ontology analysis revealed 28 upregulated and 40 downregulated miRNAs that target 123 genes experimentally observed to be associated with sirtuin signaling and senescence.

## Data Availability

The data are available from the corresponding author on request.

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
