# Peer review of "Alteration of Autophagy and Glial Activity in Nilotinib-Treated Huntington’s Disease Patients"

_metabolites, 2022, doi:10.3390/metabo12121225_

Round 1
Reviewer 1 Report
This article investigates the effect of a low dosage of nilotinib on motor and behavioural symptoms in 6 HD patients as well as the effect on HVA, neurofilaments and mutantHTT levels in CSF and plasma.
The article is novel and interesting to the HD research field.
I have a few comments and questions.
1. There is a typo on line 46 my should be may, I believe.
2. Why are the material/methods section below the results section? Is this a journal policy? It's not very logical in my opinion.
3. Why do the authors state that their primary objective is the effects of nilotinib on chorea and psychiatric symptoms, while there is only one small paragraph that describes the results of this part. Most of the results describe the analysis of the mentioned biomarkers and miRNA. However, the latest is not mentioned in the introduction.
4. To me it is not clear how the number of included patients is met. Did the authors do a power analysis? Could they please describe it?
5. The number of included patients is small, which the authors also acknowledge in their discussion. Within this group of patients there no increase of motor or psychiatric symptoms of HD was shown. However, it is impossible to conclude that this would also be the case in a larger group of HD patients. Therefore, I believe the authors should consider if they are able to answer their primary research question. I believe, they investigated if the study drug was safe and if, as they conclude in the discussion, nilotinib alternates miRNA's that may be beneficial on the breakdown of mutantHTT. But perhaps I am mistaken.
Author Response
Point-by-point response
We thank the editor for handling this manuscript and the reviewers for a thorough and insightful comments. Please see below a point-by-point response to the best of your abilities. We hope these these responses and clarifications are sufficient for the acceptance of this manuscript. Many thanks
The article is novel and interesting to the HD research field.
I have a few comments and questions.
- There is a typo on line 46 my should be may, I believe.
Corrected. Thanks
- Why are the material/methods section below the results section? Is this a journal policy? It's not very logical in my opinion.
A final corrected version is included.
- Why do the authors state that their primary objective is the effects of nilotinib on chorea and psychiatric symptoms, while there is only one small paragraph that describes the results of this part. Most of the results describe the analysis of the mentioned biomarkers and miRNA. However, the latest is not mentioned in the introduction.
We thank the reviewers for this comment. We added in the introduction line 66 in the new version, “measure HVA, phosphorylated neurofilament, HTT levels and unbiased whole genome miRNA sequencing of the CSF between baseline and 3 months.
- To me it is not clear how the number of included patients is met. Did the authors do a power analysis? Could they please describe it?
No formal power analysis was performed as this is an exploratory proof-of-concept study with a primary objective that the known effects of nilotinib on dopamine metabolism does not exacerbate behavioral symptoms (e.g. irritability) and chorea in a small population of HD. However, since this study met its primary objectives with an indication of measurable biomarkers, it is now possible to perform a power analysis for upcoming larger double blind clinical trial.
- The number of included patients is small, which the authors also acknowledge in their discussion. Within this group of patients there no increase of motor or psychiatric symptoms of HD was shown. However, it is impossible to conclude that this would also be the case in a larger group of HD patients. Therefore, I believe the authors should consider if they are able to answer their primary research question. I believe, they investigated if the study drug was safe and if, as they conclude in the discussion, nilotinib alternates miRNA's that may be beneficial on the breakdown of mutantHTT. But perhaps I am mistaken.
The reviewer is correct. This question also relates to the previous question number 4. There is no placebo group in this exploratory study, and considering that a low dose of nilotinib, 150mg, did not exacerbate Chorea and irritability in this very small cohort, a future and adequately powered placebo-controlled study should help answer the effect of nilotinib on chorea and irritability, particularly that despite the small exploratory cohort there is an indication that dopamine metabolism in significantly changed. We concur that this study has these weaknesses, but the primary objectives was to know whether 150mg nilotinib can cause a change in dopamine metabolism that leads to exacerbation of symptoms and the answer in this small cohort is encouraging, and as the manuscript suggested, a future larger study is needed. Many thanks
Reviewer 2 Report
Anderson et al.’s manuscript titled Alteration of Autophagy and Glial Activity in Nilotinib-Treated Huntington’s Disease Patients’ investigated the effect of Nilotinib HD patients via CSF miRNA screen and other metabolic analysis. This study is very straightforward, and the results are solid based on their complete analysis. However, some key issues still need to be addressed before moving to publication.
1. Give the full name of CSF
2. Remove the hyperlink in line 279
3. Can the author explain how to use ANOVA (MEANOVA) with Kruskall-wallis testing (Figure 2b) to compare/analyze baseline and 3-month DEMs.
4. The protein and mRNA level change of those autophagy genes (such as, ATG9B, ATG12, BECN1, and SESN1) between baseline and 3 months is required. Therefore, using western blot and real-time PCR to show the change is an essential validation approach. Also, the LC3 level in protein and mRNA is required to show the autophagy process change, if any.
5. Similarly, for the microglial activity analysis, please do the same validation.
Author Response
We thank the editor for handling this manuscript and the reviewers for a thorough and insightful comments. Please see below a point-by-point response to the best of your abilities. We hope these these responses and clarifications are sufficient for the acceptance of this manuscript. Many thanks
- Give the full name of CSF
Cerebrospinal fluid.
- Remove the hyperlink in line 279
Done. Thanks
- Can the author explain how to use ANOVA (MEANOVA) with Kruskall-wallis testing (Figure 2b) to compare/analyze baseline and 3-month DEMs.
We apologize for the oversight. This entire paragraph was corrected as follows:
A total of 2681 miRNA expression profiles at the baseline and 3 months were quantified in UMI counts., Longitudinal differentially expressed miRNA (DEMs) were determined via DESeq2 with ratio of means normalization and followed by Wald test [65] to compute the longitudinal measurement at baseline and 3 months. The DEMs were further evaluated using the Ingenuity Pathway Analysis (IPA, QIAGEN Inc.) microRNA Target Filter to match associated gene targets that are reported or already verified based on experimentally validated interactions from TarBase and miRecords, as well as predicted interactions from TargetScan. Gene targets were then filtered to include only highly predicted and experimentally supported mRNA target predictions. Gene targets were filtered to include only highly predicted and experimentally supported mRNA target predictions. Gene ontology (GO) pathway and functional enrichment analyses were performed on the gene targets of the significant DEMs using PANTHER Classification System [66]. Only enriched pathways and GO terms that met Benjamini-Hochberg p < 0.05 were reported.
- The protein and mRNA level change of those autophagy genes (such as, ATG9B, ATG12, BECN1, and SESN1) between baseline and 3 months is required. Therefore, using western blot and real-time PCR to show the change is an essential validation approach. Also, the LC3 level in protein and mRNA is required to show the autophagy process change, if any.
We thank the reviewer for this important point. Unfortunately, we have repeatedly tried in this study and others to detect the autophagy proteins in the CSF via both Western blot and ELISA, but these proteins are either not detectable in the CSF or no specific ELISA exist to profile them. However, as the manuscript shows, several proteins were measured and profiled to confirm miRNA changes in this cohort, including neurotransmitters (dopamine and HVA) and inflammation, HTT proteins, and the cell death marker phospho-neurofilament. We previously performed several preclinical studies as the references suggest and showed robust and consist changes of these autophagy proteins in several animal models treated with Nilotinib. Furthermore, we published the effects of Nilotinib via PCR (Fowler et la, 2020, Human Molecular Genetics) to validate the miRNA changes seen in the CSF. Most importantly, we do not have sufficient volumes of CSF to isolate and prepare mRNA for Real-time PCR, due to the amount of work and measurement performed on the samples. Therefore, we fully acknowledge the concern of this reviewer, and we plan to include real-Time PCR measurement of mRNAs in future more adequately powered and larger studies.
- Similarly, for the microglial activity analysis, please do the same validation.
Please refer to Supplemental figure 2 in which we show a panel of neuro-inflammatory protein markers via ELISA that confirm changes in microglial activity.
Round 2
Reviewer 2 Report
I have no more comments.